# Why it is important to consider negative ties when studying polarized debates: A signed network analysis of a Dutch cultural controversy on Twitter

Anna Keuchenius[1]*, Petter Törnberg[2], Justus Uitermark[2]

**1** Department of Sociology, University of Amsterdam, Amsterdam, The Netherlands, **2** Department of Geography, University of Amsterdam, Amsterdam, The Netherlands

* a.keuchenius@uva.nl

**Data Availability Statement:** The data that support the findings of this study are openly available in figshare at http://doi.org/10.21942/uva.14176493.v1.

## Abstract

Despite the prevalence of disagreement between users on social media platforms, studies of online debates typically only look at positive online interactions, represented as networks with positive ties. In this paper, we hypothesize that the systematic neglect of conflict that these network analyses induce leads to misleading results on polarized debates. We introduce an approach to bring in negative user-to-user interaction, by analyzing online debates using signed networks with positive and negative ties. We apply this approach to the Dutch Twitter debate on 'Black Pete'—an annual Dutch celebration with racist characteristics. Using a dataset of 430,000 tweets, we apply natural language processing and machine learning to identify: (i) users' stance in the debate; and (ii) whether the interaction between users is positive (supportive) or negative (antagonistic). Comparing the resulting signed network with its unsigned counterpart, the retweet network, we find that traditional unsigned approaches distort debates by conflating conflict with indifference, and that the inclusion of negative ties changes and enriches our understanding of coalitions and division within the debate. Our analysis reveals that some groups are attacking each other, while others rather seem to be located in fragmented Twitter spaces. Our approach identifies new network positions of individuals that correspond to roles in the debate, such as leaders and scapegoats. These findings show that representing the polarity of user interactions as signs of ties in networks substantively changes the conclusions drawn from polarized social media activity, which has important implications for various fields studying online debates using network analysis.

## Introduction

In recent years, the advent of social media platforms has given researchers access to a wealth of digital data on social relations, behavior, and beliefs [1, 2]. Data from these social media platforms have fueled the growth of new approaches for social research, most notably

**Funding:** This research has received funding from the European Union's Horizon 2020 research and innovation program under grant agreement No 732942, project ODYCCEUS.

**Competing interests:** The authors have declared that no competing interests exist.

Computational Social Science, which uses digital data and computational methods to capture and study social dynamics [1–3]. Drawing from the natural and technical sciences, Computational Social Science provides powerful tools and methods for working with large-scale relational data, opening up new avenues into the study of social phenomena such as mass mobilization [4, 5], polarization [6, 7], the spread of misinformation [8], political discourse [9], and much more. In this research, social network analysis is among the most powerful and commonly used tools; by representing social interaction as graphs, the network perspective unearths the relational structures emanating from and shaping interactions, allowing researchers to identify communities and central actors [2].

Network studies into polarization have shown that online users sharing ideological affiliation tend to cluster together in terms of interaction [10–12], which suggests an understanding of polarization as the simultaneous clustering of allies and repulsion between antagonists [13, 14]. However, while animosity and conflict are central to this relational angle on polarization, the vast majority of social network studies into polarized online debates have only considered positive relationships, represented as networks with positive ties [15, 16]. This has resulted in confounding results, such as finding cohesive network structures in what are known to be polarized debates [10, 11, 17, 18]. This paper argues that the systematic neglect of negative, antagonistic, user-to-user interactions in network studies has severe consequences for our understanding of polarized discourse online. We introduce an approach, powered by natural language processing and machine learning, for distinguishing positive from negative interactions between users. This information on the polarity of user interactions allows for the analysis of online debates using a signed network, a network with positive and negative ties. Our analysis shows that the inclusion of negative ties has a profound impact on the findings with respect to the structure of the debate and the positions of actors and communities within it.

We apply this approach to a case study on the divisive Dutch debate over 'Black Pete' ('Zwarte Piet'), a Dutch mythical character with racist connotations. Although black communities and anti-racist activists have long critiqued Black Pete, in recent years, a full-fledged national debate has unfolded about the character and what it might say about racism in Dutch society more broadly [19, 20]. This debate provides a useful case to show how signed network analysis enhances our understanding of contentious debates.

We use a dataset of the Twitter debate on Black Pete, covering the period from December 2017 to May 2019, comprising roughly 430,000 tweets from 81,700 unique users, with 296,881 unique mentions between users. From this dataset, 10,000 tweets were manually labeled, coding their issue sentiment (*pro / neutral / anti / ambiguous* in relation to the issue of Black Pete), and the sentiment of each user-to-user interaction (*opposition / agreement / neutral / ambiguous* in relation to the targeted user). Using this labeled dataset, we trained a machine learning algorithm to classify the interaction sentiment between users. From that, we constructed a signed network of users in this debate and compared this to the retweet network for the same data, which is typically employed to study polarization on Twitter. The comparison shows that our approach identifies a larger number of actors, finds different communities, and provides greater insight into the diversity of roles that actors play within the debate. In particular, we show important differences between actors that are attacked from all sides (mainly cabinet members and public institutions) and actors that receive support from one side while coming under attack from the other side (mainly parliamentarians and activists).

In the remainder of this introduction, we outline advances in computational Twitter research on polarization and signed network analysis. In the materials and methods section, we first present our case study and dataset on Black Pete and subsequently describe our methods for extracting the sign of user relations. The result section is composed of four subsections. The first two empirical subsections compare the signed network with the retweet network in

terms of 1) users included in the network, and 2) the structure and composition of communities. The latter two empirical subsections detail the relations between communities and the roles of individuals in the debate that are laid bare by the (positive and) negative interactions. These results are powered by quantitative as well as qualitative analyses of the data in order to reach meaningful conclusions about their significance. Finally, we conclude what our approach to signed network analysis contributes to the study of polarization.

## Literature: Twitter studies and signed networks

To capture the structure of interaction, Twitter research has focused on the social networks that are shaped by user interaction through either "retweets" or "mentions," both of which are generally studied through unsigned network analysis. A retweet is a simple act of sharing in which a user shares another user's tweet with their network. Retweets are generally considered endorsements, that is, as positive ties [21] or as contributing to information flow [22]. When studying debates through retweet networks, researchers have found separate user clusters, with limited interaction between political opponents [10, 11, 17, 23–26]. The correlation between retweet structure and political ideology is so strong that retweet networks have been used to predict user ideology [11, 17, 25, 27].

Mentions, on the other hand, are a syntax for targeting a message or part of a message to a specific user, by adding the @-sign to the Twitter username. For example, a user in our dataset mentioned Nadia Bouras, a historian that publicly speaks out against Black Pete with over 29,000 followers on Twitter, in the following tweet: '@NadiaBouras Cry Baby. Black pete stays anyway!!!'. In previous literature, mentions are generally considered neutral, or as expressions of information exchange. Conover et al. [11], for instance, suggest that "mentions form a communication bridge across which information flows between ideologically-opposed users". When studying debates through mention networks, researchers generally do not find strict divides between opposing groups, as mentions occur across polarized clusters and party lines [10, 11, 17, 28–30], concluding that information exchange is occurring across political lines (33).

However, the reliance on unsigned ties in studying Twitter debates means that both retweet and mention network representations have important limitations. Whereas research based on retweet networks ignores interactions across clusters through mentions, research on mentions inadvertently conflates positive and negative interactions. As the example above illustrates, mentions can be used to attack other users, instead of as a tool to share information with them. This has also been found in more qualitative studies on Twitter. Evolvi [31], for instance, studied Islamophobic tweets in the aftermath of Brexit, and found that mentions are often used to "belittle others with different ideas rather than invite conversation" (p. 396). Similarly, Moernaut, Mast and Temerman [32] studied polarized climate discourse on Twitter and found that interactions tend to be antagonistic, aimed at delegitimizing and denaturalizing out-groups. Gruzd and Roy [28] manually labeled tweets that occur across party lines and found that roughly half of these are hostile. This would suggest that previous studies have conflated very different forms of interaction, mischaracterizing out-group derogation as a form of neutral information flow. Recent computational research that models social group formation via signed networks further suggests that the exclusion of negative ties significantly distorts community structure [33]. The neglect of negative interaction thus appears to have severely limited the capacity of network analysis to accurately represent online debates, with important implications for the many fields relying on this approach.

To date, there is limited research on Twitter debates using signed networks. This is in part due to the difficulty with identifying the polarity of mentions. This is not a simple variable in

the data but has to be abstracted from the meaning of the words and position of the mention in the tweet. One approach to identifying the sign of ties between users that has been applied in previous literature is to simply assume that the interaction of users that hold opposite positions will always be negative, while interactions among users with the same position will always be positive [34, 35]. Another approach taken is to focus on online social networks that allow for explicit negative relations between users, such as Epinion or Slashdot. Studies using such data, however, have predominantly been aimed to develop algorithms to predict signs of edges or future link creation rather than answering social scientific questions about polarization or other social processes (for an overview of signed network mining see [36]).

When signed networks of online data have been studied in relation to social processes, they have typically been used to test theories on social balance [37] and status theory [15, 38–42]. There are examples of studies using signed network representations of offline data to research polarization. Neal [43] examined the level of polarization in US congress by representing co-sponsorship of bills as a signed network of interactions between congress members. Uitermark et al. [44] investigated the Dutch debate on minority integration in newspapers through signed network analysis, demonstrating that opposing groups' community structures differ in terms of cohesion and leadership. Traag and Bruggeman [45] studied international alliances and disputes, establishing the world is divided into six power blocks. While these studies demonstrate the importance and scientific potential of using signed networks, they also illustrate the challenges involved in extracting signed networks from debate data. Scholars either manually classify relations, use niche social platforms, or make strong assumptions on the sign of ties–all of which preclude the use of signed networks in the study of mainstream social media platforms like Twitter. This paper presents a method for moving beyond this impasse by automatically extracting the polarity from online user interaction in large-scale social media debates by using natural language processing and machine learning.

## Materials and methods

### Case: Is Black Pete racist?

The celebration of Sinterklaas (Saint Nicholas) is one of the most important traditions of the Netherlands [46]. Saint Nicholas is similar to Santa Claus: he has a long white beard, a red outfit, and he brings presents for children. Saint Nicholas arrives by steamship from Spain every year in early November and is welcomed publicly in almost every Dutch city. A single town is nominated to be the host of the official national welcoming of Sinterklaas, which means having the occasion broadcasted on national television. On the evening of Saint Nicholas, the 5th of December, the Saint visits families across the country, presenting gifts and sweets to children. In the days and weeks leading up to the 5th of December, many shops are decorated with Sinterklaas-themed promotional material.

The part of this tradition that has become an issue of contention are the helpers who accompany Sinterklaas: the "Black Petes" ("Zwarte Pieten"). These are usually represented by white people wearing blackface. The character has periodically become the focus of debate in Dutch society, due to their—for most observers from outside the Netherlands rather striking —racist undertone [46–48]. The current wave of debate started with the arrest of four activists, most notably Quinsy Gario and Jerry Afriyie, for their participation in protests against Black Pete in Dordrecht in 2011 during the official welcoming [46, 49]. Since this protest, there have been intense debates in newspapers, on television, in parliament and up to the UN on whether Black Pete embodies a racist stereotype [49].

This debate intensified from 2013 onwards, with supporters and opponents of Black Pete mobilized online and in the street every year, leading to violent confrontations. In 2017, pro-

Black Pete activists blocked a highway in the north of the Netherlands to prevent anti-Black Pete activists from protesting at the official welcoming of Saint Nicholas. The debate about Black Pete has become the focal point in broader debates about Dutch racism, Dutch colonialism, and the Netherlands' involvement in the transatlantic slave trade [48, 49]. Opponents believe that Black Pete exemplifies Dutch racism whereas opponents see Black Pete as an innocent character and consider criticisms as an attack on their traditions by overdemanding minority groups and arrogant cultural elites [46–48].

## Data

We used a dataset of tweets on the Black Pete debate posted between December 4th, 2017 and May 7th, 2019. The tweets were collected based on keyword matching of various terms related to the debate, such as "Black Pete", "Zwarte Piet" and "KOZP" (abbreviation for "Kick Out Zwarte Piet"), harvested and stored using the Twitter Capture Analysis Toolset [50]. In total, the dataset contains 467,497 unique tweets from 81,700 unique users, with 296,881 unique mentions between users.

## Ethics statement

The data collection process has been carried out exclusively through the Twitter API, which is publicly available, and for the analysis, we used publicly available data (users with privacy restrictions are not included in the dataset). We abided by the terms, conditions, and privacy policies of Twitter. Since this content is publicly published and is frequently discussed in mass media, we regard the debates as a public domain that does not require individual consent for inclusion in research, based on the ethical guidelines for internet research provided by The Association of Internet Researchers [51] and by the British Sociological Association [52]. We only report on aggregates, and limit reporting on details of individuals to user accounts that belong to public figures or institutions, or that have more than 4,000 followers. The data published along with this research does not include user-ids nor the classification of the sentiment on the Black Pete discussion since this is part of a special category of personal data, formerly known as sensitive data.

## Issue and mention sentiment classification

To classify the relationships between users (positive, neutral, negative), we identified, for each tweet (1) the *issue sentiment*–the position expressed on the issue of Black Pete, and (2) the *mention sentiment*–the position toward each mentioned user in the tweets, i.e., whether the tweeting user mentions the other user to express agreement, opposition, or is neutral, such as sharing information. It should be noted that we did not try to classify the overall sentiment of the tweet, for which various existing sentiment analysis algorithms could be deployed, but we specifically targeted the position of the user in relation to Black Pete and the sentiment of the interaction with the mentioned user.

To infer these sentiments, we first manually classified approximately 10,000 tweets randomly selected from the full dataset. By this selection method, we avoid focusing on the most active or popular users which limits the bias towards the vocal minority to the detriment of the (more) silent majority [53]. We coded the issue sentiment, whether the tweet expresses a pro, anti, or neutral/ambiguous stance towards Black Pete, as well as the sentiment of each mention in each tweet. We took into consideration that one tweet might contain several mentions, some of which might be intended positively towards the mentioned users and others might be signaling disagreement. These labeling efforts were conducted by four fluent Dutch speakers who were instructed via a coding book designed for this project. The codebook instructions

were conservative: if the issue or mention sentiment was not self-evident, the tweet was labeled as ambiguous (see the S1 Appendix for more details). The inter-coder agreement was moderate to substantial, measured by a Krippendorf Alpha of 0.72 for the issue sentiment and 0.49 for the mention sentiment, indicating that the classification is a difficult task.

To classify the rest of the data, we applied the following pipeline. First, we classified the issue sentiment of all tweets by the fastText algorithm [54] trained on the manually labeled issue sentiments (see the S1 Appendix for more details). Second, we count the number of *pro* and *anti*-tweets of each user and categorized users' stance as pro- or anti-Black Pete by a simple majority rule. That is, if the user posted more pro than anti tweets, we assigned a pro label to the user, and vice versa. Third, we trained the fastText algorithm to classify mention sentiments using the manually labeled mention data. In addition to the tweet text, we provided the fastText model with information about the issue sentiment of the tweet as well as the stance of the tweeting and mentioned users (classified in the previous steps). We additionally constructed two features that might reveal information about the sentiment of the mention: (1) whether the mention takes the form of "via @username"—which are most often neutral, as they are automatically added by the webserver of the media outlet via which the tweet was posted—and (2) whether the mention is located at the start, body, or end of the tweet since that might correlate with the polarity of the mention.

For both the classification of issue sentiment and mention sentiment, we implemented the fastText algorithm for text classification [54], which is informed by advances in word representation learning [55, 56]. This algorithm uses the training data to construct numerical word vectors for each word in the corpus that represents their relation to other words, thereby capturing (part of) their meaning. To teach the model the basics of the Dutch language, we provided the model word vectors constructed from a Dutch Wikipedia Corpus [57]. The use of such an external corpus enables the machine learning algorithm to discover similarities in words that are missing or infrequent in its training data, thus increasing its vocabulary and subsequent predictive power.

Since the manually labeled data included many more tweets that support than oppose Black Pete (58% expressed a pro position), we balanced the class sizes before classifying the issue sentiment to avoid biasing the algorithm. The fastText classifier categorizes the issue sentiments with sufficiently high accuracy, resulting in 15% (65.314) anti tweets, 48% (225.856) pro tweets and 38% (176.327) tweets with neutral/ambiguous issue sentiment (see the S1 Appendix for more details on the issue sentiment classification). Similarly, the labeled mentions were not balanced, containing more negative than positive user mentions. We down-sampled the majority classes to avoid biasing the algorithm, resulting in 1,382 positively annotated mentions, 1,500 negatively annotated mentions and 1,500 neutral/ambiguous mentions. For the mention classification, we filtered the test data on unique tweet text to ensure that the test data included only tweet texts that the classifier had not seen before. We did not filter the training set on unique tweet texts to ensure the classifier learned that one tweet can include several mentions with different mention sentiments.

Since we aim to identify positive and negative mentions, we optimized the algorithm to minimize the risk of incorrectly classifying a negative tweet as positive, and of classifying a positive tweet as negative. We are less concerned with incorrectly classifying a positive or negative tweet as neutral since this will have less impact on distorting the resulting network. To do this, we trained the classifier to maximize the F1 score for all classes, thus attempting to predict all classes well, in both precision and recall. The fastText algorithm gives an indication of how certain the classification is (the softmax probability), valued between 0 and 1 for each prediction. We used this certainty indication to apply a simple rule: classify all mentions with lower certainty (<0.8) as neutral. This procedure reduces the recall for the positive and negative

classes, but more importantly, reduces the errors we care most about: classifying positive mentions as negative, and classifying negative mentions as positive.

The classifier—after applying the certainty rule—categorizes the mention sentiments with high accuracy (see Fig 1). There are only 21 cases in which a negative mention is misclassified as positive (0.031 times of all negative mentions and of 0.15 times all positive mention classifications) and 22 cases in which a positive mention is misclassified as negative (0.07 times of all positive mentions and 0.064 times of all negative classifications). To classify user-to-user interaction signs, we considered both the mentions and retweets, where retweets are taken as acts of endorsements, a positive interaction from the retweeting to the retweeted user [58]. Next, we used a majority rule: if most of the user-to-user interactions were positive (negative), we classified the directed sign between these users as positive (negative). This procedure classified approximately 54% of user interactions as neutral (or too ambiguous to categorize), 37% as positive and 8% as negative.

Since we conducted a signed network analysis, we focus on relations that we could with some certainty identify as positive or negative with the procedures described above, while leaving out neutral and ambiguous relations. Most (86%) of the neutral/ambiguous relations were based on only one interaction between the users and were therefore more difficult to classify accurately.

## Results

The classification of the sentiment of user interactions (positive, negative) allows us to construct a signed network of this debate and compare that network to the retweet network that is commonly used for studying Twitter debates. In our comparison of the signed network with the retweet network, we focus on (1) differences in the set of users included in both networks; (2) the overall community structure and composition in the networks; (3) the positions of these communities in the network and the debate; and (4) the role of individual actors in the network and debate.

The signed network consists of 94,016 nodes (representing users), 150,555 positive and 33,329 negative relations. The retweet network, which is based on direct retweets ("quote tweets" are treated as mentions), consists of 55,758 nodes with 211,669 relations. We consider

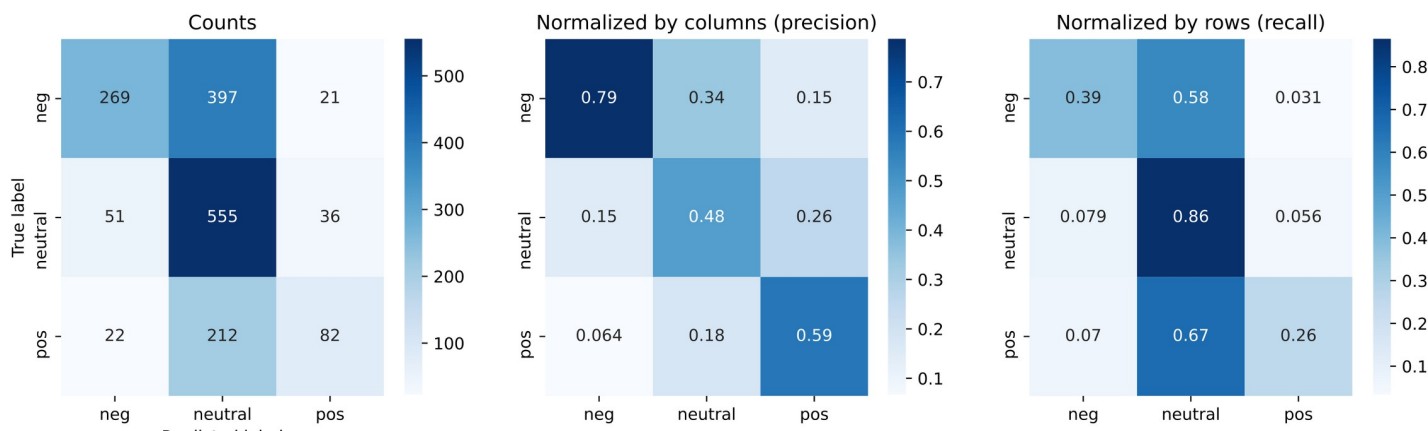

**Fig 1. Confusion matrix mention sentiment.** The results of the classifier (parameter values: epoch = 25, learning rate = 0.7, n-grams = 3) after applying the simple certainty rule (neutral if certainty < 0.8): confusion matrix with counts (left), normalized by the true labels (middle) and normalized by the predicted labels (right). The values in the diagonals of the middle matrix are the precision rates, and the values on the diagonals of the right matrix are the recall rates. Recall rates here are reduced due to the certainty rule, but the most important errors (classify positive if the true value is negative and classify negative if the true value is positive) are reduced.

the edges in the retweet network as positive, in line with previous research with which we aim to compare our signed network results [10, 17, eg. 24, 26, 58].

## Missing users

The first notable difference between the retweet network and the signed network is that they include different actors. In total, there are 38,258 (40%) users in the signed network that are not in the retweet network. These are users that are not being retweeted (because they are not tweeting on the topic in our data) but are receiving mentions on Twitter in the context of the debate on Black Piet. However, many of these users are isolates or not part of the largest connected component of the signed network. We, therefore, focus our subsequent analysis on the largest connected component of each network, as is typically done in network analysis. There are 3,112 users (6%) in the signed network that are not present in the retweet network. In comparison, 559 users (1%) in the retweet network are not part of the signed network. The users that an analysis of the retweet network misses out on tend to be more important in the debate; the users missing in the signed network have very low (all less than 50) indegree, whereas many of the users the retweet network misses out on are prominent in the debate (see Fig 2).

Taking a closer look at users that have a high indegree in the signed network but are missing in the retweet network (see Fig 3), we find that these users are key actors in the debate on Black Pete. For example, the prime minister of the Netherlands Mark Rutte ('minpres' on Twitter) is absent in the retweet network as he did not tweet about the topic. However, his words and actions in this debate are influential and many people mention him on Twitter, giving him a central position in the signed network. Similarly, the public prosecutor (referred to by users as @om) is often mentioned negatively but is absent in the retweet network as this account did not tweet on the topic.

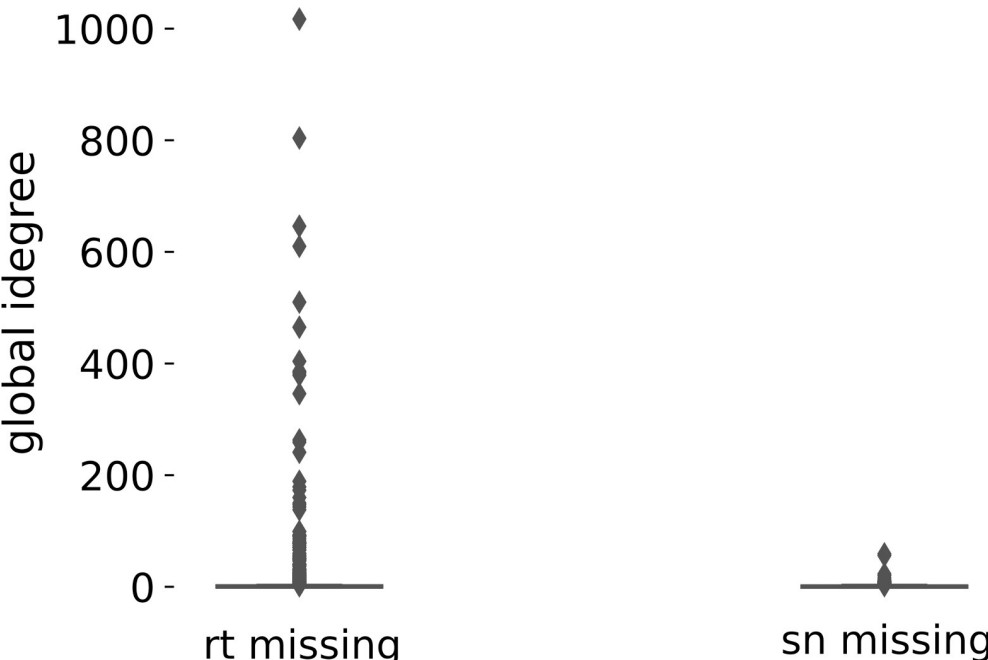

**Fig 2. Missing users in retweet versus signed network.** Distribution of the indegree of the users that an analysis of the largest connected component in the retweet network (left) or signed network (right) would miss.

| user_name | followers | community | global indegree | pos global indegree | neg global indegree |
|---|---|---|---|---|---|
| jesseklaver | 264,000 | 4 | 1017 | 11 | 1006 |
| minpres | 1,000,000 | 4 | 804 | 4 | 800 |
| omroepntr | 43,700 | 2 | 646 | 8 | 638 |
| sylvanasimons | 12,700 | 2 | 610 | 4 | 606 |
| om | 1,300,000 | 4 | 510 | 0 | 510 |
| albertheijn | 45,600 | 4 | 465 | 0 | 465 |
| kruidvat | 505 | 2 | 404 | 1 | 403 |
| publiekeomroep | 45,200 | 4 | 386 | 0 | 386 |
| politie | 271,300 | 4 | 383 | 34 | 349 |
| kruidvatservice | 7,750 | 2 | 379 | 1 | 378 |

**Fig 3. Top missing users.** The top 10 users of the signed network that are not in the retweet network. These top ten users are not in the retweet network at all—also not in the smaller or isolated communities. The column statistics are based on the signed network. The follower counts are by July 2020.

## Community structure

We next compare the structure and composition of communities in the signed network with those in the retweet network. We detect the community structures in both networks with the Leiden Algorithm [59], which maximizes the positive links within communities and minimizes positive links between communities compared to a random network with the same degree distribution. In the case of the signed network, the negative edges are also taken into account but with the reverse logic: minimizing the number of negative edges within communities and maximizing the number of negative edges between communities [45].

Both networks show similar degrees of modularity: the retweet network has a modularity of 0.45 while the signed network has a modularity of 0.45 in the layer of positive edges and of 0.24 in the layer of negative edges. In both networks, the two largest groups are of a similar size and contain roughly 40% of the nodes, and the largest ten communities together make up roughly 90% of the nodes (see Fig 4).

Whereas the community structure in both networks are similar in modularity and size, the communities differ significantly in terms of their compositions. Comparing the users and communities of the retweet network directly with those of the signed network reveals that the mentions and their polarity have had a marked influence on the group compositions (see Fig 5). For example, the users classified in community 1 in the signed network, the dominant pro-Pete community, overlap for a substantial part (66%, n = 7,784) with the users in the same community in the retweet network. However, they are merged with many other users (n = 3,854) from other communities in the retweet network, such as community 5 and 6. At the same time, several users belonging to community 1 in the retweet network are split off into other separate communities in the signed network, community 4 and particularly community 7. This community 7 is a community centered around Geert Wilders, a radical right-wing politician with a strong anti-immigration and anti-Islam agenda.

Zooming in on the twenty most prominent actors (those who receive the highest number of retweets, positive or negative mentions), we find that some of these top users are grouped in different combinations in the retweet network compared to the signed network, illustrated in Fig 6. In sum, taking into account mentions with their signs affects the composition of communities for both rank-and-file as well as prominent actors.

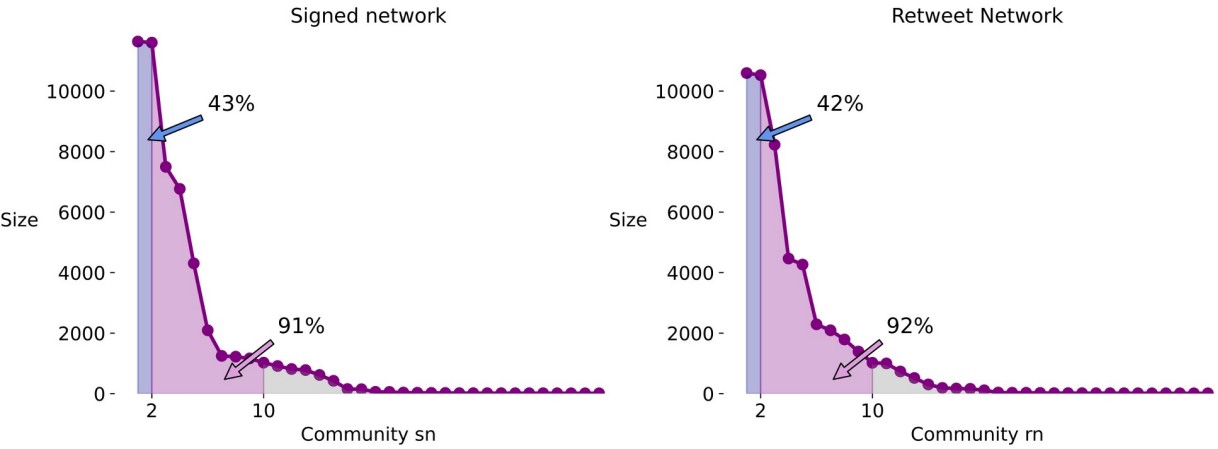

**Fig 4. Community sizes.** The distribution of sizes of the communities in the signed network (left) and retweet network (right). The tail is cut-off (displaying only communities with more than 10 members) for the sake of legibility. This figure shows that the two networks feature a similar community structure.

## Coalitions and divisions in the debate

After exploring the differences in community structure and composition between the two networks, we turn to what the negative interactions contribute to the understanding of these

## Community Differences

**Retweet network**                **Signed network**

**Fig 5. Community differences.** Alluvial graph illustrating the relationships between the group structure in the retweet (left) and signed network (right). The thickness of the lines correspond to the number of users, and non-horizontal lines indicate differences between the group structures in the two networks. The figure shows considerable differences in the group compositions and illustrates that there are many central users in the signed network (n = 3.112) that are not in the top 18 communities of the retweet network.

communities' positions in the debate and their relations with one another. Whereas the communities in the retweet network are formed on the basis of separation, the signed network detects groups on the basis of separation and confrontation which leads to richer information on the community relations and their positions in the debate. Fig 7 displays the relationships between the signed network communities in terms of relative positive ties (left) and negative ties (right) and illustrates each communities' dominant stance towards Black Pete (pro/anti/neutral) by a color scale. This shows that some communities send many negative messages to others in the debate, even to communities with a similar aggregate pro/anti-stance on this polarizing topic.

The next section provides a more detailed interpretation of these findings. This is based on the community statistics reported in Fig 8, relations between communities in the network visualized in Fig 7, and the users' stance on Black Pete (see Fig 9). We then carry out a qualitative analysis of the main communities in the signed network (those with over 1,000 members), analyzing their twenty most retweeted tweets, all tweets of the community's users with the highest positive indegree, and a random sample (n = 100) of other tweets of the community. For this selection of tweets, we examine the themes addressed and the position expressed toward Black Pete.

The signed network is dominated by two large, antagonistic poles: one constituted by pro-Black Pete communities 1 and 4, and another by the anti-Black Pete community 2 (see Fig 9). Community 1 of the pro-Black Pete pole is most vocal (users on average positively referencing almost 9 others in their community) and most confrontational (attacking on average 1.7 users of other groups) (see Fig 4). There are other outspoken pro- and anti-Black Pete communities,

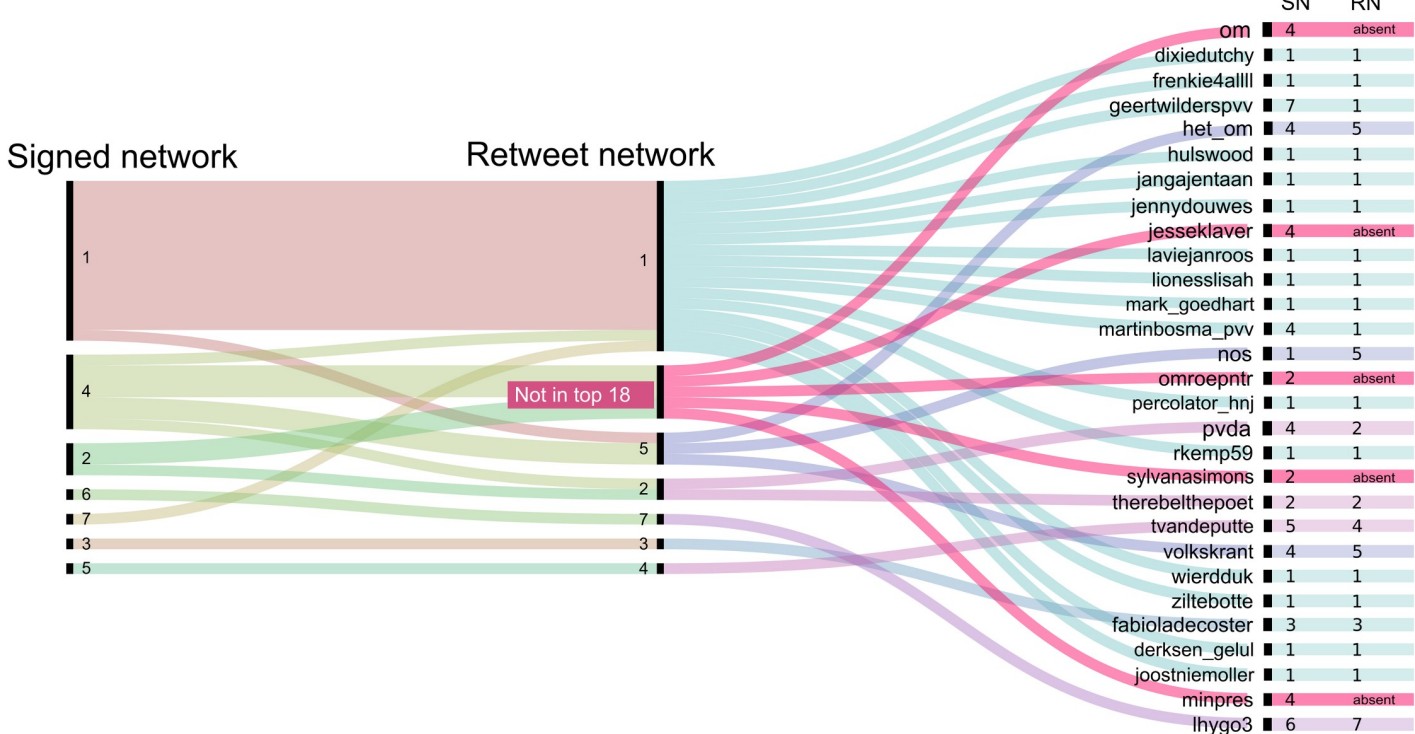

**Fig 6. Community differences top actors.** Alluvial graph illustrating the communities of the top 20 actors in the debate in the signed (left) versus retweet network (right). The figure shows considerable differences in the group structures and illustrates that some of these top actors in the debate, such as prime minister Mark Rutte (minpres), are not present in the top 18 communities of the retweet network.

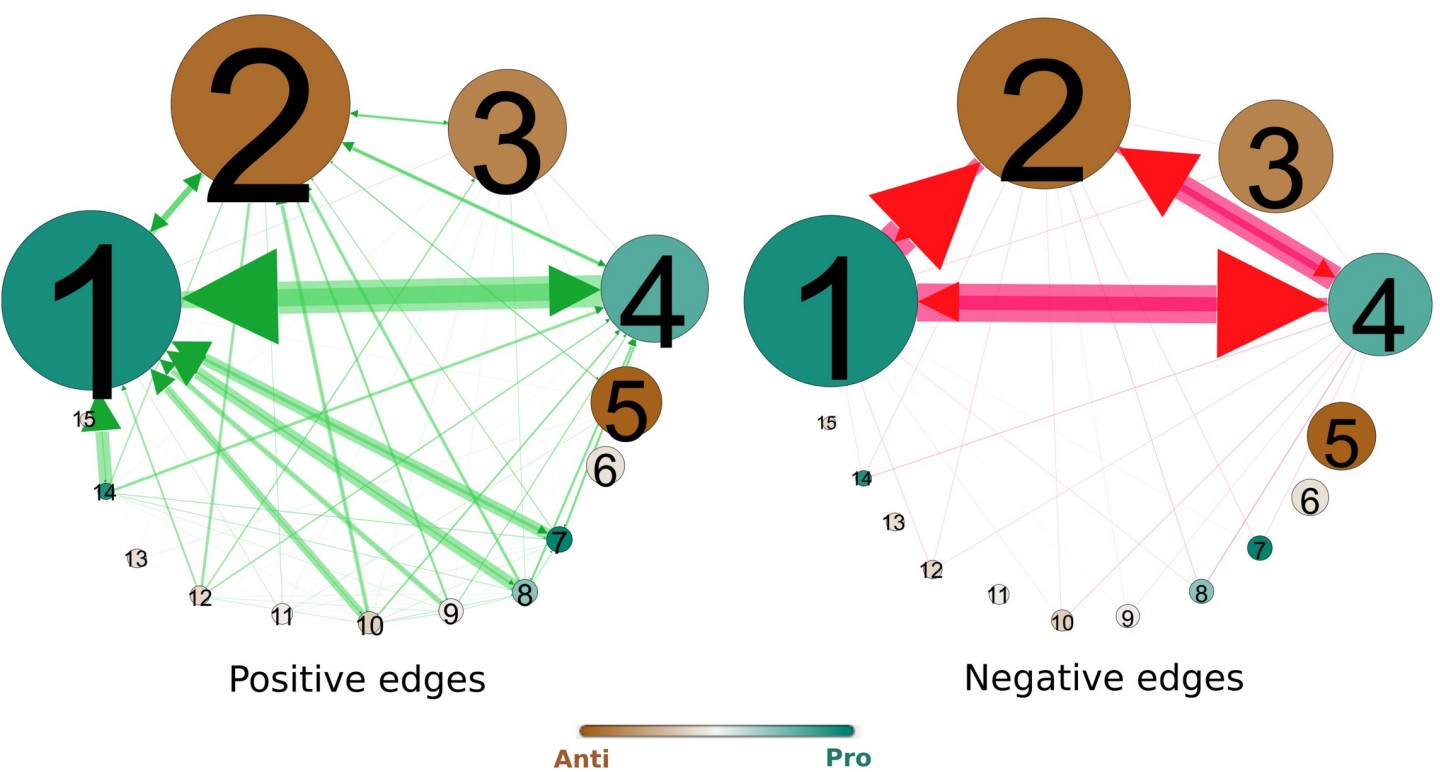

**Fig 7. Community relations.** The aggregate network of the communities, with positive relations (left) and negative relations (right). Nodes are sized by the number of users in this community and colored by the average issue sentiment of users in the community. Edges are sized by the absolute count of outgoing edges from source to target community, divided by the source community's size.

such as community 7 and 3 respectively, but there is significantly less antagonistic communication from and to these communities (see Fig 7). Furthermore, some communities (9 and 10) feature pro as well as anti-Black Pete users, together averaging to a neutral position on Black Pete. Popular tweets express exasperation with the debate.

*Community 1*, the vocal and confrontational pro-Black Pete community, is one of the largest communities in the network with roughly 11,000 users. Users in this community show strong internal cohesion (users positively reference almost 9 others in the community on average) and heavily attack users from other communities (negatively referencing 1.7 users of other communities on average). Users of this community are vehemently pro-Black Pete and mainly attack users in the anti-Black Pete community 2. This community's stars are Joost Niemoller and Wierd Duk, both journalists and well-known pro-Black Pete supporters. Other prominent figures are the anonymous Twitter account @perculator_hjn (which produces a stream of tweets expressing radical right opinions) and Jenny Douwes, the initiator of a road barricade to block anti-Black Pete protesters in 2017. The main targets of attack are the anti-Black Pete activist Jerry Afriyie (@therebelthepoet, community 2), who often gets scolded for his activism and is told that he should "go back to Ghana," and the public prosecutor (@het_om, community 4) that is accused of being biased against supporters of Black Pete.

*Community 2*, the activist anti-Black Pete community, also includes roughly 11,000 users and is the main antagonistic pole of community 1. Users in this community tend to hold anti-Black Pete positions and include many of the core anti-Black Pete activists, as well as politicians, newspapers, and national celebrities that have spoken out in favor of changing the appearance of Black Pete. This community is also internally cohesive and externally negative,

| community | size | avg issue sentiment | avg issue sent top users | pos int e | avg pos int e | density (pos) | pos int e fr. | neg out e | neg out e fr. | avg neg ext e | indegree pos | indegree neg | top local positive | top global negative |
|---|---|---|---|---|---|---|---|---|---|---|---|---|---|---|
| 1 | 11,638 | 0.77 | 1.00 | 104,011 | 8.94 | 0.04 | 0.87 | 19,337 | 0.94 | 1.66 | 15,418 | 5,936 | joostniemoller, wierdduk, percolator_hnj, jennydouwes, hulswood | nos, wierdduk, teletekst, eenvandaag, tponl |
| 2 | 11,603 | -0.57 | -1.00 | 35,044 | 3.02 | 0.01 | 0.84 | 5,738 | 0.94 | 0.49 | 6,158 | 13,252 | therebelthepoet, vicenl, thenuc1, nadiabouras, claricegargard | therebelthepoet, omroepntr, sylvanasimons, erikvmuiswinkel, kruidvat |
| 3 | 7,496 | -0.46 | -0.70 | 11,435 | 1.53 | 0.01 | 0.91 | 38 | 1.00 | 0.01 | 1,015 | 70 | fabioladecoster, anomalisa__, aranxerini, rani_remy, queentrey_ | ninodevries, 2020gwoww, mickjohan, queentrey_, ronnieflex2907 |
| 4 | 6,768 | 0.58 | 0.70 | 8,663 | 1.28 | 0.01 | 0.42 | 6,103 | 0.94 | 0.90 | 10,877 | 11,789 | martinbosma_pw, franckentheo, asterberghe, bartdemeulenaer, robinvanhijfte | het_om, jesseklaver, minpres, volkskrant, pvda |
| 5 | 4,306 | -0.63 | -0.80 | 4,660 | 1.08 | 0.01 | 0.95 | 4 | 1.00 | 0.00 | 622 | 24 | tvandeputte, karenattiah, marcusjdl, tinalasisi, soualiganamazon | karenattiah, tvandeputte, kelechnekoff, redlightvoices, solitudekth |
| 6 | 2,094 | 0.00 | 0.00 | 2,097 | 1.00 | 0.02 | 1.00 | 0 | 0.00 | 0.00 | 1 | 0 | lhygo3, lookslikechloe, seongbukrotari, cjsanseo1110, ena_hand_real | lhygo3, wdbb0u39, ena_hand_real, poroli_love, ssc_for_running |
| 7 | 1,248 | 0.84 | 0.50 | 1,298 | 1.04 | 0.04 | 0.60 | 10 | 1.00 | 0.01 | 2,184 | 32 | geertwilderspvv, voetbalflitsen, denieuwemaanntr, voetbalinside, defendevropa | geertwilderspvv, denieuwemaanntr, supercemal, voetbalflitsen, saabje96 |
| 8 | 1,222 | 0.42 | 0.70 | 1,394 | 1.14 | 0.05 | 0.57 | 17 | 1.00 | 0.01 | 1,500 | 93 | danevv, rtlnieuws, adnl, pro_respect, johan_anema | rtlnieuws, adnl, hartvnl, nhnieuws, rtvnoord |
| 9 | 1,165 | 0.06 | 0.20 | 1,198 | 1.03 | 0.04 | 0.70 | 4 | 1.00 | 0.00 | 538 | 28 | haralddoornbos, ozcanakyol, evoosterhout | ozcanakyol, haralddoornbos, drbrambakker |
| 10 | 1,025 | -0.08 | 0.00 | 1,086 | 1.06 | 0.05 | 0.61 | 11 | 1.00 | 0.01 | 1,003 | 16 | peter26061980, rloppenheimer, sylviawitteman | sylviawitteman, vice, mariannezw |
| 11 | 913 | 0.07 | 0.10 | 926 | 1.01 | 0.06 | 0.93 | 0 | 0.00 | 0.00 | 158 | 9 | ajenglish, redfishstream, rt_com | ajenglish, breakingnlive, rt_com |
| 12 | 815 | -0.03 | 0.40 | 1,072 | 1.32 | 0.08 | 0.78 | 10 | 1.00 | 0.01 | 234 | 15 | lectrr, youssef_kobo, philippedecoene | pensioenspook, vanranstmarc, elhammouchiothm |
| 13 | 780 | -0.00 | 0.00 | 780 | 1.00 | 0.06 | 0.99 | 0 | 0.00 | 0.00 | 3 | 0 | onifinau, madameghana, ajplusfrancais | madameghana, zzaynabtata, adaugo__ |
| 14 | 617 | 0.75 | 0.70 | 647 | 1.05 | 0.09 | 0.55 | 10 | 1.00 | 0.02 | 656 | 11 | jankoudum, bigbolder, maxdekok1913 | rob_rtm, jankoudum, omropfryslan |

**Fig 8. Community statistics signed network.** Community statistics and their top users in the signed network (with size>100). The average issue sentiment is calculated over the sentiment (pro-Black Pete = 1, anti-Black Pete = -1) of all users, and reported separately for the top 10 users most often positively related to from within the community. Column 'pos int e fraction' divides the positive edges within the community by the total positive edges outgoing from community members and 'neg out e fr.' divides negative edges within the community by the total negative edges outgoing from community members. The columns 'avg pos int e' and 'density (pos)' divide the total positive edges by the number of users and the possible edges in the community, respectively. The column 'top global negative' lists the users in this community that are most frequently negatively mentioned by other communities.

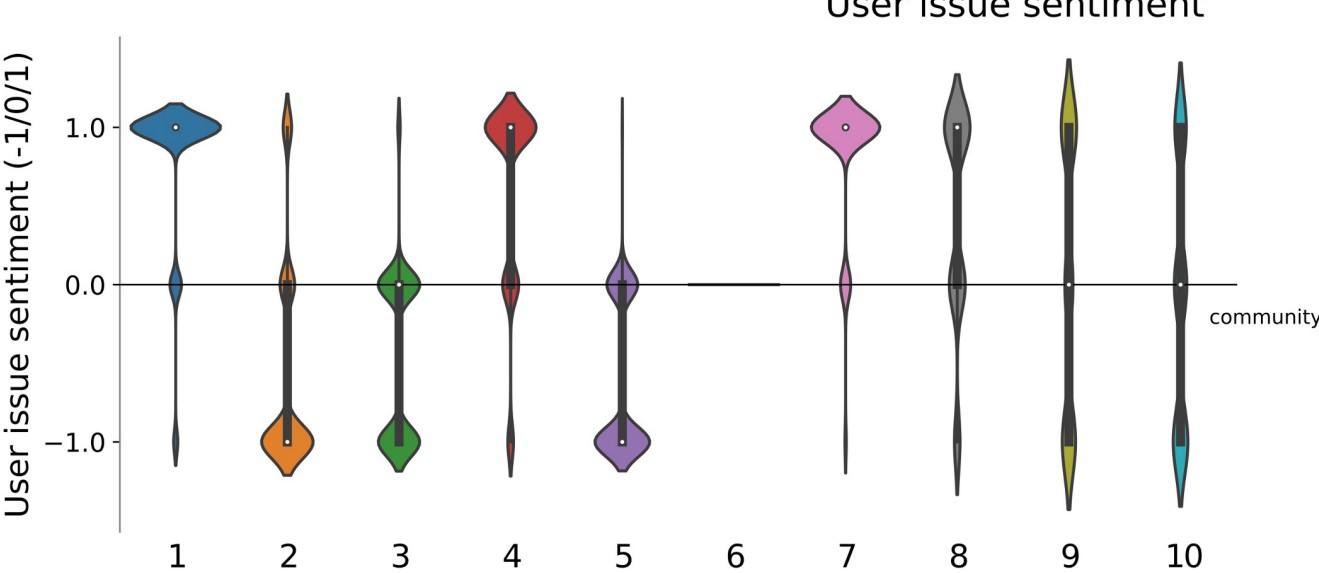

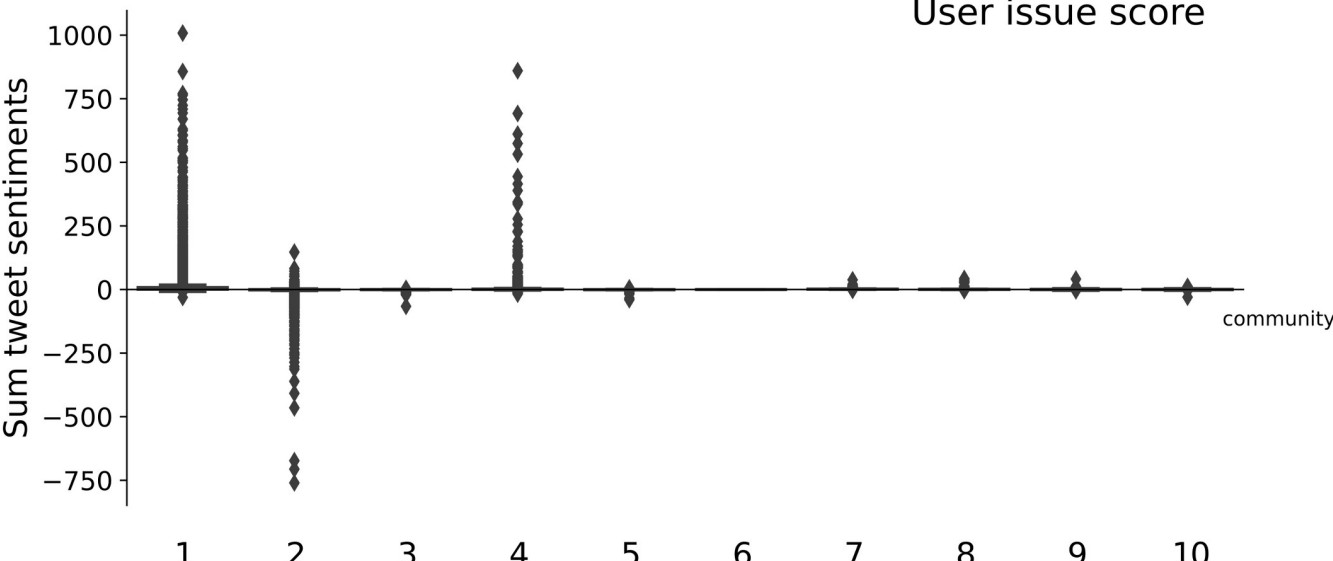

**Fig 9. Distribution communities' issue sentiment.** The distribution of users' issue sentiment in favor (1) or against (-1) Black Pete per community. The top panel gives the final user issue sentiment by the majority rule (-1,0 or 1 per user). The bottom panel gives the users' issue score, calculated by the sum of users' tweet sentiments. This shows communities 1, 2 and 4 contain a number of highly active users with a strong sentiment on Black Pete.

though less pronounced than the pro-Pete community 1. The majority (62%) of negative references of community 2 are directed towards users of community 1, followed by users of community 4, the second pillar of the pro-Black Pete pole (30%). Community 2's central figures are Jerry Afriyie (@therebelthepoet); ViceNL, a media outlet; the New Urban Collective (@NUC1), an activist social enterprise for inclusivity; and Nadia Bouras (@nadiabouras), a historian working for Leiden University. The main targets of attack are NOS (@nos, community 4), the Dutch Broadcasting Foundation (comparable to the BBC); Wierd Duk (@wierdduk, community 1); and the Dutch Prime Minister (@minpres, community 4).

*Community 3* has 7,496 users and is an anti-Black Pete community that stands out for its relatively young and international members. This community is mostly formed around positive internal relations instead of negative external relations. Tweets by these users often feature slang, and particularly English slang ("y'all", "wanna", "trash", etc.) and are often more jovial, for instance discussing the Black Pete issue in relation to dating. Most (75%) of the positive references to other communities are directed towards the activist anti-Black Pete community 2.

*Community 4* is overwhelmingly pro-Black Pete and consists of 6,768 users. These users are identified as a community predominantly because of criticisms they direct at others (0.9 per user on average) and that others direct at them (1.75 per user on average). Yet there are also positive connections within the community (1.28 per user on average). This community is mostly in opposition to the activist anti-Black Pete community 2. The relationship with the vocal and confrontational pro-Pete community 1 is more ambiguous: users of community 4 reference community 1 positively as well as negatively, and similarly, users of community 1 reference them positively as well as negatively. This community includes many institutions and institutional actors, such as the public prosecutor (@het_om), the police (@politie), some political parties, and municipalities. These accounts tend to predominantly be subject of negative links from other users. Internal positive edges are centered around one politician, Martin Bosma (@martinbosma_pvv), who is part of the radical right-wing party of Geert Wilders, the PVV. Bosma is very active in the debate, retweeting over 40 distinct users with pro-Black Pete tweets.

*Community 5*, with 4,306 users, is another anti-Black Pete community that is internationally oriented (most of the tweets are in English) but is older and more academic than community 3. The community is structured around positive internal edges (1.08 per user on average), more so than outgoing or receiving negative critique. The community is organized around Tom van de Putte (@tvandeputte), a Dutch academic who is head of the critical studies department at the Sandberg Institute in Amsterdam the Netherlands. He is retweeted 2,066 times in this community (out of his 5,199 retweets).

*Community 6*, with 2,094 users, is an exceptionally isolated community whose users neither retweet nor mention users from other communities, and receive in total only one reference from a user in another community. This community predominantly tweets in Korean (98%) and uses no mentions. The community is centered around tweets from one user account (@lhygo3) that has been suspended, but who tweeted both in Korean and English about their surprise about the existence of Black Pete in the Netherlands.

*Community 7*, the Wilders community, has 1,248 users and is outspoken pro-Black Pete. Users are centered around tweets of Geert Wilders, positively referenced by 63% of them, who uses the Black Pete issue to explicitly call upon users to vote for his political party (PVV). The community is formed around positive internal edges (1.04 per user on average), and less so by negative incoming or outgoing critiques. This community has a remarkably high number of positive incoming edges from other pro-Black Pete users, predominantly from the vocal and confrontational pro-Black Pete community 1 (76%), and to a lesser extent from community 4 (13%). Reciprocally, users in this Wilders community 7 also positively reference users from these two pro-Black Pete communities.

*Community 8*, the media community, with 1,222 users, receives many positive and negative references from other communities relative to its size (1.30 per user on average), particularly from the activist anti-Black Pete community 2. Many of the top accounts in this community are from news outlets, such as television shows, radio broadcasters and newspapers. The majority of tweets by users in this community express a pro-Black Pete position.

*Community 9*, one of the neutral communities, consists of 1,165 users with solely positive relations between them. Unlike many other communities, this community is not centered

around one highly retweeted or mentioned user. The tweets in this community are not expressively pro-Black Pete, but frame the discussion as irrelevant, making jokes and comparisons with other issues they deem irrelevant. The mayor of the city Emmen, for example, tweets that he would rather deal with creating job opportunities than with issues such as Black Pete, fireworks during New Year's, or some other controversial symbolic political issues in the Netherlands.

*Community 10* is another neutral community and has 1,025 users. This community is, similarly to community 9, not expressively anti-Black Pete but its members emphasize they are exasperated by what they see as an overblown discussion. About a third of this community's positive outgoing edges are directed to other communities, most frequently to the vocal and confrontational pro-Black Pete community 1 as well as the activist anti-Black Pete community 2. Users of this neutral community are also positively referenced by both of these communities.

This examination has shown that the inclusion of negative ties not only changes the composition of communities but also reveals a more complex structure of internal fractions and coalitions within and between the supporters and critics of Black Pete. Compared to retweet networks, signed networks enables distinguishing conflict from indifference, which creates a richer understanding of the community structure. For instance, while communities 1, 4 and 7 are all predominantly pro-Black Pete, there is a high level of negative interaction between communities 1 and 4. Between the anti-Black Pete communities, we see few negative interactions. It seems as if users of these communities are subsiding in segregated Twitter spaces, not regularly mentioning, or retweeting each other. This important difference would have been impossible to discover using traditional unsigned network analysis.

## Signed structural positions and debate roles

Unsigned networks have two widely recognized structural positions: *hubs* (users with many ties) [60, 61] and *bridges* (users that connect otherwise separate communities) [62]. These are widely used in the study of social networks. In the study of political debates, these positions are taken to correspond to roles taken by actors: hubs are central actors or opinion leaders [63], and bridges play the role of mediators between communities. However, since unsigned networks do not consider the nature of the interaction, these structural positions can be argued to map poorly to roles in polarized debates. Using unsigned networks implies either considering only positive interaction, thereby missing actors that have important roles as subjects of critique, or conflating positive and negative interaction, thereby confusing venerated authorities with hated trolls.

Using signed networks, we can identify a larger number of structural roles, since a given node can be important in terms of negative ties or positive ties, for members of one side of the debate, the other side, or both sides. The argument made here is that these structural network positions more directly map to roles in the debate, by allowing to distinguish popularity from infamy. This provides a central network tool for the examination of polarized debates. We here identify structural positions and their corresponding roles, using the Black Pete debate network for illustrative purposes.

The spectrum of structural positions in a debate with two opposing sides can be represented as a matrix with two axes: on the x-axis, there is the sentiment of one side (operationalized as the number of positive ties minus the number of negative ties) and on the y-axis, there is the sentiment from the other side of the debate. This two-dimensional landscape produces a typology of structural positions based on different regions in the matrix (see Fig 10). We identify five different positions and their corresponding roles:

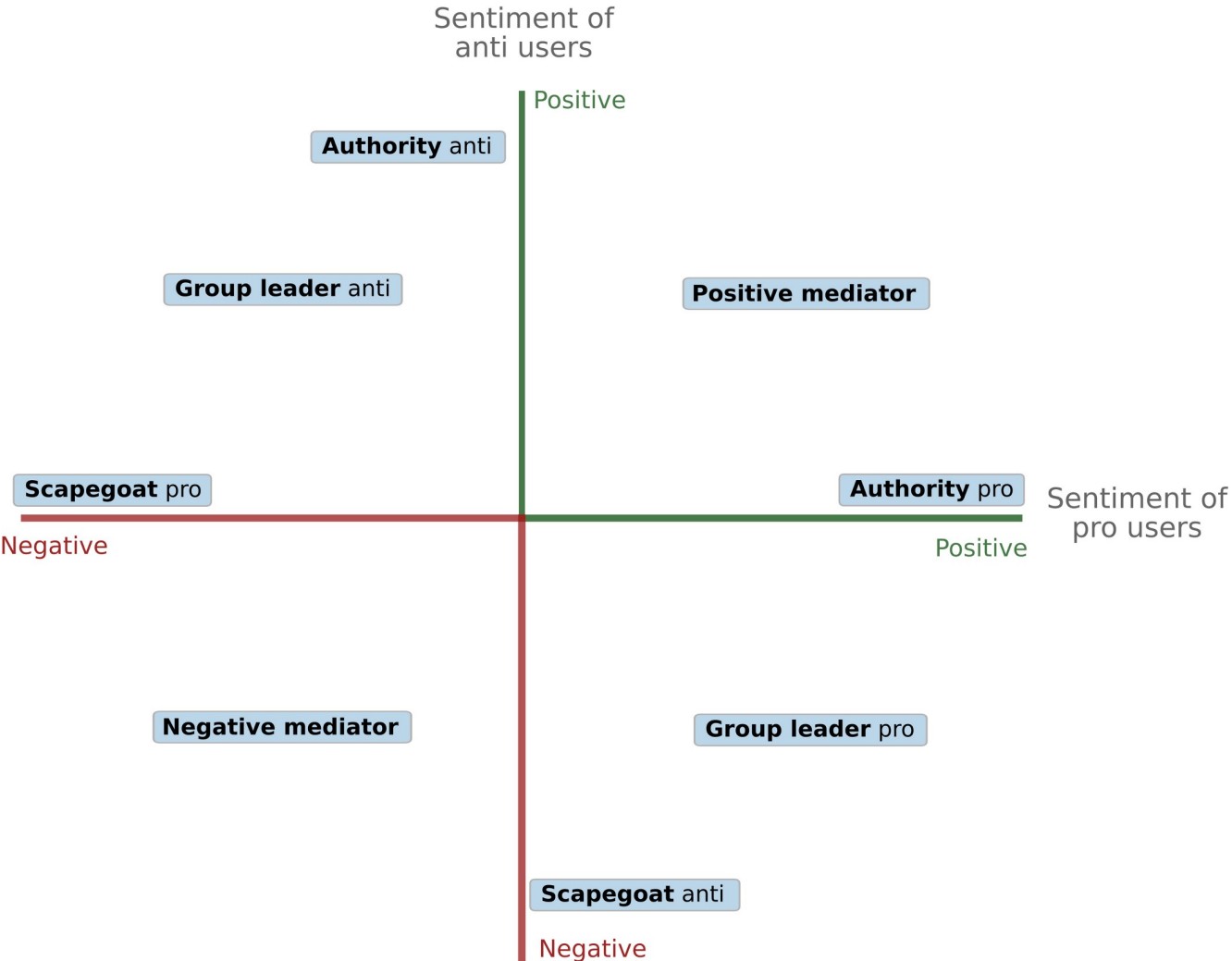

**Fig 10. User typology in sentiment landscape.** Theoretical landscape of positions in the debate, as defined by the way users of both sides (pro and anti) reference the users. The x-axis and y-axis represent the average sentiment of pro- and anti-users, respectively.

- *Group leaders* receive many positive references from users on their side of the debate, implying that they are recognized as representatives of their cause, and many negative ties from the other side of the debate, implying that the opposing group also views them as important representatives.

- *Group authorities* also receive a lot of positive references by users on their side of the debate but are not attacked by users on the other side, perhaps because they are seen as poor representatives or targets for attacks.

- *Scapegoats* are strongly negatively referenced by the opposing group but ignored or neutrally referenced by users of the side to which they belong. Scapegoats tend to be users that are seen as useful targets of attacks for the opposing group, representing aspects of their outgroup that activates their group solidarity but are not considered as leaders by their ingroup.

- *Positive mediators* are referenced positively by both sides of this debate. Due to this position, positive mediators may function to reduce tensions between the groups.

• *Negative mediators* are referenced negatively by both sides of the debate, though not necessarily for the same reasons. By being the object of dislike from both groups, they potentially bring the groups together by constituting a form of common ground [64].

Fig 11 shows the structural position landscape of the Black Pete debate, with the users that are most often referenced annotated and colored by their respective communities. The figure identifies the main *group leaders* of both sides: Jerry Afriyie (@therebelthepoet) on the anti-side, and Wierd Duk (@wierdduk) on the pro-side. Both figures are considered by their opponents as radicals who fire up their base to attack the other side.

In the lower-left quadrant in the figure, we see Twitter profiles of *negative mediators* that are heavily attacked by both sides of the debate, most notably NOS (@nos), the Dutch Broadcasting Foundation, which is criticized by both pro and anti-users with claims of biased reporting. Other negative mediators are also predominantly institutions, politicians, and media, such

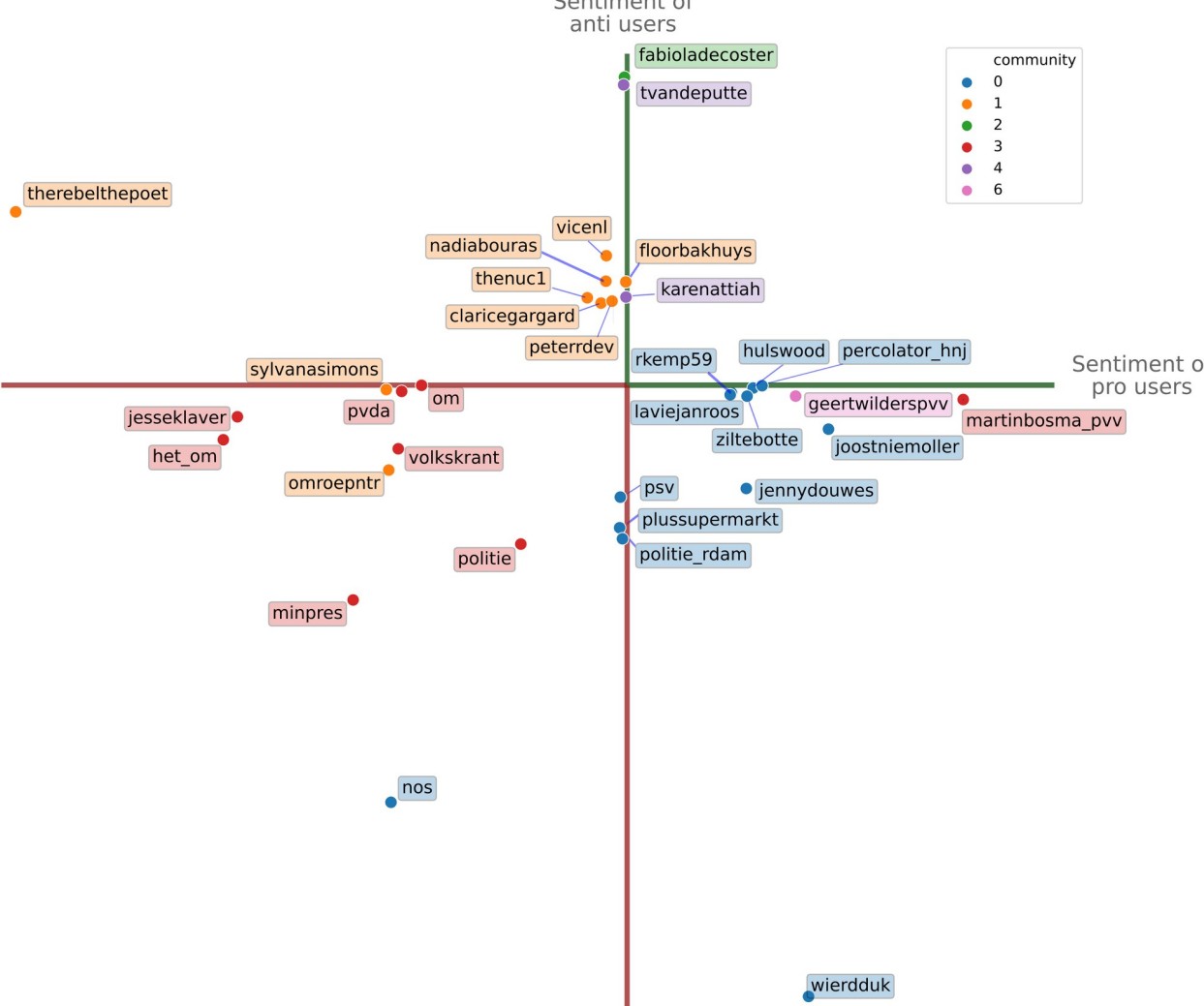

**Fig 11. User typology of the Black Pete debate.** Positions of top users in the Twitter Black Pete debate, as defined by the way users of both sides (pro and anti) reference the users. The x-axis and y-axis represent the average sentiment of pro and anti-users respectively, in which the positive (negative) number of edges to each user is normalized by the total positive (negative) edges of pro (for x-axis) and anti (for y-axis) users in total. Users are colored by their respective communities in the signed network.

the police (@politie), the prime minister Mark Rutte, the Dutch national television station that broadcasts the children's show on the celebration (@omroepntr), the public prosecutor (@het_om), and one of the main national newspapers (@volkskrant).

On the left side of the horizontal axis, we find two actors with *scapegoat* positions, receiving many negative mentions from pro-users, but few positive mentions from anti-users: Sylvana Simons (@sylvanasimons) and the social-democratic political party PvdA (@pvda). Sylvana Simons is a politician and founder of the anti-racist party Bij1, who has previously been subject to racist threats and hateful attacks. At the lower end of the vertical axis, we find the corresponding scapegoats for anti-users: the football club PSV (@psv) whose supporters allegedly intimidated anti-Black Pete protesters; the supermarket Plus (@plussupermarkt), which—unlike other supermarkets—did not ban the Black Pete characters from products and came under scrutiny for having White employees dressed as Black Pete in its stores; the city of Rotterdam police department (@politie_rdam) which is accused of violating the right of protest of opponents of Black Pete. The scapegoat users thus tend to be institutional actors, often without an official position in the debate, that are targeted as symbolic for the bias of mainstream actors, but lack important discursive roles for the side to which they are taken to belong.

*Group authorities* are the mirror image of *scapegoats*: they are positively referenced from their side of the debate but receive no or only neutral references from the opposing side. The most important authorities on the pro-side are @percolator_hnj, @hulswood and @rkemp59. These are not public figures such as politicians, journalists, or institutions, but instead are activists on Twitter who have nonetheless built a large following (19,000, 11,000 and 10,000 Twitter followers, respectively). On the anti-side, the most influential accounts are @fabiola-decoster (4,000 followers) and @tvandeputte. De Coster does not tweet in a formal capacity; Tom van de Putte is Head of Critical Studies at the Sandberg Institute.

These examples show how the structural positions in signed networks correspond to differentiated social roles in the debate that would not be possible to identify using unsigned network analysis. For example, a highly attacked scapegoat of one side would be missed by a retweet network, or perhaps worse, would be taken as popular figures for the other side by a mention network. The variety of roles in this signed analysis is much broader than can be grasped when negative ties are not taken into account.

## Conclusion and discussion

Twitter has become a central data source for the rapidly growing research on social phenomena using digital data [65, 66]. Data on debates on Twitter have been used to deepen our understanding of a range of phenomena, including mass mobilization [4, 5], polarization [6, 7], the spread of misinformation [8], political discourse [9], and much more. One of the most central methodological pillars underlying this research is the use of social networks to represent interactions between individuals in debates [2].

However, while it is self-evident that in the study of polarized debates it is necessary to distinguish conflict from indifference, leadership from pariahdom, information sharing from insults–doing so has nonetheless been impossible using classical social network methods because these include only one type of interaction: positive interaction. This paper has presented an approach for addressing this limitation by extracting the polarity (positive, negative) of user interaction online and subsequently analyzing the debate using a signed network representation (with positive and negative ties). We applied this approach to Twitter data on the polarized Dutch debate around 'Black Pete,' an annual tradition that has become a lightning rod for the country's culture wars. By processing the tweets on this issue using natural

language processing and machine learning, we detect the polarity of user mentions, which we use to extract a signed network of user interaction.

By comparing the resulting signed network with the commonly used unsigned retweet network, the paper showed that signed networks allow for a substantially richer understanding of online debates. First, the signed network captures important and influential users that are missing in the retweet network. Second, the user composition of the identified communities in the signed network differs significantly from the unsigned retweet network. Third, signed networks allow for the identification of not only separate but also conflicting fractions. Our analysis showed that some groups are attacking each other, while others seem to be located in fragmented Twitter spaces–an important distinction that would be impossible to make using unsigned analysis. Fourth, signed networks allow us to distinguish a greater variety of structural positions, which better correspond to roles taken by actors in the debate. Rather than only *hub* and *bridge*, we identified five roles in the debate: *leaders*, *authorities*, *scapegoats*, *positive mediators*, and *negative mediators*.

This shows an important flaw in the existing approaches to studying debates on Twitter and other social media through unsigned networks. These networks with only positive ties systematically neglect or misinterpret negative, antagonistic, sometimes hostile user interactions. We have shown that some of the directed messages to other users (through mentions) do not constitute a "flow of information" (13), but are rather expressions of antagonism, contention and disagreement of the type that sociologists have long argued are central to the process of group formation. These findings have implications for a broad range of research using social media data, suggesting that research needs to begin considering the sign of the interaction when employing network representations of debates.

The primary limitation of the approach introduced in this study is that it requires a labeled set of training data to use supervised machine learning to detect the interaction sentiment in tweets. In contrast to other popular machine learning classification tasks, such as sentiment detection, there are currently no pre-trained classifiers or training data available. However, future research might provide such resources. As this study has focused on a specific debate, embedded in a specific time period, country, and social media platform, future research may study whether the identified patterns hold more broadly, by expanding its approach to study group structures and intergroup communication online in a variety of political debates, countries and platforms. Future research may also focus on what this signed network representation can tell us of the dynamics of political polarization in social media, by shifting our understanding of online polarization from isolation and fragmentation to conflict and confrontation.

## Supporting information

**S1 Appendix.**
(DOCX)

## Author Contributions

**Conceptualization:** Anna Keuchenius, Petter Törnberg, Justus Uitermark.

**Data curation:** Anna Keuchenius.

**Formal analysis:** Anna Keuchenius.

**Funding acquisition:** Petter Törnberg, Justus Uitermark.

**Investigation:** Anna Keuchenius.

**Methodology:** Anna Keuchenius, Petter Törnberg, Justus Uitermark.

**Project administration:** Anna Keuchenius.

**Resources:** Anna Keuchenius.

**Software:** Anna Keuchenius.

**Supervision:** Petter Törnberg, Justus Uitermark.

**Validation:** Anna Keuchenius.

**Visualization:** Anna Keuchenius.

**Writing – original draft:** Anna Keuchenius, Petter Törnberg, Justus Uitermark.

**Writing – review & editing:** Anna Keuchenius, Petter Törnberg, Justus Uitermark.

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
