## [Decision Letter · Decision Letter 0]

26 Apr 2021

PONE-D-21-08059

Why it is important to consider negative ties when studying polarized debates on Twitter

PLOS ONE

Dear Dr. Keuchenius,

Thank you for submitting your manuscript to PLOS ONE. After careful consideration, we feel that it has merit but does not fully meet PLOS ONE’s publication criteria as it currently stands. Therefore, we invite you to submit a revised version of the manuscript that addresses the points raised during the review process.

The paper needs a MAJOR REVISION. Please, revise the manuscript in order to follow the PLOS publication guidelines, and improve the quality of the paper by following the lacks highlighted by the reviewers.

We look forward to receiving your revised manuscript.

Kind regards,

Barbara Guidi

Academic Editor

PLOS ONE

Journal Requirements:

2. Please consider changing the title so as to meet our title format requirement (https://journals.plos.org/plosone/s/submission-guidelines). In particular, the title should be "Specific, descriptive, concise, and comprehensible to readers outside the field" and in this case it is not informative and specific about your study's scope and methodology.

3. Please include a copy of Table 1 which you refer to in your text on page 8.

Reviewers' comments:

Reviewer's Responses to Questions

**Comments to the Author**

1. Is the manuscript technically sound, and do the data support the conclusions?

Reviewer #1: Yes

Reviewer #2: Partly

Reviewer #3: Partly

2. Has the statistical analysis been performed appropriately and rigorously? 

Reviewer #1: Yes

Reviewer #2: No

Reviewer #3: N/A

3. Have the authors made all data underlying the findings in their manuscript fully available?

Reviewer #1: No

Reviewer #2: No

Reviewer #3: No

4. Is the manuscript presented in an intelligible fashion and written in standard English?

Reviewer #1: Yes

Reviewer #2: No

Reviewer #3: Yes

5. Review Comments to the Author

Reviewer #1: I thank the authors for this work that I am certain will be of interest to many given the subject matter of the debate and the authors' application of signed network analysis. For me the detail and nuance in the community definitions was the most appealing and informative part of this paper. I think there is some value in seeing the nature of participants and how they participate in this or other debates.

I base my recommendation on potential to improve the presentation of content and organization, primarily in the introductory section of the paper. Following I provide some specific comments:

According to the PLOS publication guidelines, the Introduction is a required paper element and it precedes the “Materials and Methods” (Methods) section. To follow this guideline, I suggest the authors combine the current Introduction with the sections that follow and precede methods – perhaps by converting the headings, e.g., “Studying debates on Twitter” etc. to subheadings. I suggest the information about methods and results shown in the introduction as paragraphs two and three should be removed, so the introduction section is focused on providing the necessary background and review of literature to contextualize and rationalize the study. I also suggest the authors consider revising the phrase “The structure of our argument” to refer to this manuscript, i.e., “This paper is organized as follows” or “In the remainder of this paper, we begin by outlining…” I do not think “argument” is the best term for an empirical research report.

Introduction – “Roughly one-third of social media users” – this is clearly stated in the cited report. I am not certain the other part of the sentence - “and social media has become one of the most important sources for learning about current political debates” - is supported in this report as broadly as the authors suggest. The cited report only speaks to social media users, so to me me this does not support “has become one of the most important sources” unless this is conditioned upon something like for instance, being a politically engaged social media used. I suggest the authors modify this statement as appropriate.

I have a similar response to the next sentence – I suggest the authors define, specify or offer a citation for “have become an important resource for social science research.” It is the assertion “important resource” that I question.

First paragraph in “Studying” section, second sentence: “These data have fueled” rather than “has”

Second paragraph in “Studying” section: “Twitter, in particular, has been the preferred social media platform” – this assertion requires a citation. It is the word “preferred” that I think needs support.

Same paragraph – it might be worthwhile to define “unsigned network analysis” if the intended audience is any social science researcher versus data scientists. Understanding this concept is important for readers to make sense of this work.

Top of third page describing prior research (Evolvi, Moernaut et al., Roy) – I believe prior research should be described using past, not present tense (Evolvi studied … and found…). This recommendation (use of past tense) also applies to other descriptions of prior research shown on this page.

“Case” section: Santa Claus (no e) is the usual spelling for the character name, even though there is a film called "The Santa Clause."

“Case” section , paragraphs two and three: I recommend the authors provide citations to support the information about protests and activism. These might come from popular or news media or any other sources consulted by the authors to prepare this report.

Methods section, paragraph four: Please describe to readers how the codebook used was designed and assessed.

Same section, paragraph five – please use past tense consistently in describing the research efforts (“counted,” “trained,” “provided,” “constructed”). This comment also applies to other sections of the methods that were written in present rather than past tense.

Reviewer #2: This is a review for “Why it is important to consider negative ties when studying polarized debates on Twitter”. The article used a dataset of the Twitter debate on Black Pete, covering the period from December 2017 to May 2019. In general, the manuscript would benefit from streamlining the overall structure and focusing the argument more tightly. Right now, the reader has to jump around to different concepts and methods, making the argument hard to follow. Focusing each paragraph on a clear topic sentence of sorts – “the main point” – and then unfolding any relevant citations and explanation in support would be useful, making sure that these citations are clearly relevant and tied to said topic sentence.

My main concerns are as follows:

1. Past tense should be paid attention in illustrating results.

2. The summarized results could be given in the abstract to attract readers’ interests and attention.

3. “Contention” as the keyword showed just once in this manuscript, which could be taken place by “trust building”.

4. Definitions of polarized debates could be given out in second paragraph, since first paragraph was the background information. Given that this is the central construct under investigation in this study, the author should make sure that it is clearly and consistently defined and operationalized throughout the manuscript. Furthermore, the author should provide clear justification, via citations of previous research or otherwise, for why their definition is the most relevant and useful one to employ.

5. What is the drawbacks of considering only positive relationships. Giving the importance of introducing an approach for identifying positive and negative interactions in online debates. “However, while animosity…has considered only positive relationships” One sentence cannot explain the gap in the previous literature. The author need to present that the current research can fill the research gap of each previous literature.

6. All the variables or instruments should be discussed in the introduction part. More importantly, review on variables of trust building items should be presented in the first part rather than in the discussion part.

7. The introduction need more elaboration. I suggest rewriting these parts.

8. Since the author mentioned social media, the effects of social bots could be considered. There are currently a handful of studies that analyze the effects of social bots. Most of these are within politically charged online conversations, including the United Kingdom’s Brexit referendum (Howard & Kollanyi, 2016), the ongoing Ukraine–Russia conflict (Hegelich & Janetzko, 2016), and the 2016 United States Presidential Election (Bessi & Ferrara, 2016).

9. How to classify the issue sentiment of all tweets by the fastText algorithm?

10. Please list the serial number and name below the figures.

11. Potential multicollinearity problems should be tested with tolerance statistics.

12. Some of the discussion of existing literature does not always seem relevant. This may be due to the argument getting muddled, or it may be that the authors can simply cut less relevant tangents.

4. The discussion should be reorganized more logically. For example, the content of "Limitations and future research direction" could be added.

MINOR POINTS:

Some references list authors completely, others do not.

The manuscript should be checked for APA style. Some grammatical errors making text hard to follow. Throughout the manuscript, there are a lot of language issues (e.g., article use of a and the, syntax). I would recommend having the paper proof read.

References:

Bessi, A., & Ferrara, E. (2016). Social Bots Distort the 2016 US Presidential Election Online Discussion. Retrieved from https:// firstmonday.org/article/view/7090/5653

Hegelich, S., & Janetzko, D. (2016). Are social bots on Twitter political actors? Empirical evidence from a Ukrainian social botnet. In ICWSM (pp. 579–582). Retrieved from https://www. aaai.org/ocs/index.php/ICWSM/ICWSM16/paper/down- load/13015/12793

Howard, P. N., & Kollanyi, B. (2016). Bots, #StrongerIn, and #Brexit: Computational propaganda during the UK-EU referendum. Retrieved from https://papers.ssrn.com/sol3/ papers.cfm?abstract_id=2798311

Reviewer #3: Thanks authors for investigating such an important subject; I have the following comments:

There are some unnecessary details in the introduction such as an elaborate explanation about black Pete. I recommend summarizing the information about the debate and instead focusing on the method that is used to analyze such a debate and its comparison with previous methods.

The Materials and method section is mixed with the Results, Discussion and Conclusions section. The materials and method section needs to be separated from other parts; only the Results, Discussion and Conclusions section can be combined.

Like the introduction, the method section has some unnecessary information and its combination with results and discussion makes it hard to follow the important points.

Figure 8 is not clear to read.

6. PLOS authors have the option to publish the peer review history of their article (what does this mean?). If published, this will include your full peer review and any attached files.

Reviewer #1: **Yes: **Sheryl L. Chatfield

Reviewer #2: No

Reviewer #3: No

---

## [Author Response · Author response to Decision Letter 0]

28 Jun 2021

Dear Barbara Guidi,

Thank you for considering our paper and for giving us the opportunity to address the issues raised by the reviewers and yourself.

You suggested we reconsider the title. We have now renamed the article to ‘Why it is important to consider negative ties when studying polarized debates: a signed network analysis of a Dutch cultural controversy on Twitter’ and hope that you agree this new title meets PLOS ONE’s title guidelines. 

We wish to thank the reviewers for their close reading of the paper and their constructive suggestions. We have now revised the paper to address the reviewers’ concerns and we believe the structure and argument of the paper is much clearer. In the document "Response to reviewers", we explain how we addressed the major issues raised by the three reviewers, shown below on a point-by-point basis. 

We have taken up the suggestion of all the reviewers to restructure the text and tighten the argument. We are grateful for this suggestion, because we believe that the article is now easier to read and brings out the main argument more clearly. As suggested by all reviewers, we have also had a professional proofreading service go through the article.

We hope you agree that our revisions adequately address the concerns that the reviewers raised. 

Kind regards,

The authors

---

## [Decision Letter · Decision Letter 1]

16 Jul 2021

PONE-D-21-08059R1

Why it is important to consider negative ties when studying polarized debates: a signed network analysis of a Dutch cultural controversy on Twitter

PLOS ONE

Dear Dr. Keuchenius,

Thank you for submitting your manuscript to PLOS ONE. After careful consideration, we feel that it has merit but does not fully meet PLOS ONE’s publication criteria as it currently stands. Therefore, we invite you to submit a revised version of the manuscript that addresses the points raised during the review process.

The paper needs a MINOR REVISION. The authors should address the issues highlighted by the reviewers concerning typos and more details about the fastText algorithm.

We look forward to receiving your revised manuscript.

Kind regards,

Barbara Guidi

Academic Editor

PLOS ONE

Journal Requirements:

Reviewers' comments:

Reviewer's Responses to Questions

**Comments to the Author**

1. If the authors have adequately addressed your comments raised in a previous round of review and you feel that this manuscript is now acceptable for publication, you may indicate that here to bypass the “Comments to the Author” section, enter your conflict of interest statement in the “Confidential to Editor” section, and submit your "Accept" recommendation.

Reviewer #1: (No Response)

Reviewer #2: All comments have been addressed

Reviewer #3: All comments have been addressed

2. Is the manuscript technically sound, and do the data support the conclusions?

Reviewer #1: Yes

Reviewer #2: Partly

Reviewer #3: Yes

3. Has the statistical analysis been performed appropriately and rigorously? 

Reviewer #1: Yes

Reviewer #2: No

Reviewer #3: Yes

4. Have the authors made all data underlying the findings in their manuscript fully available?

Reviewer #1: Yes

Reviewer #2: No

Reviewer #3: No

5. Is the manuscript presented in an intelligible fashion and written in standard English?

Reviewer #1: Yes

Reviewer #2: Yes

Reviewer #3: Yes

6. Review Comments to the Author

Reviewer #1: Thanks to the authors for their work to revise this manuscript. I appreciate their efforts to edit and to provide thorough and thoughtful responses to authors. I found this version more concise and engaging. I have one very small comment/recommendation - in the introduction I think there may be a minor grammatical error. The sentence readers "Drawing the natural and technical sciences..." Should this be "Drawing on the natural and technical sciences" or perhaps "Drawing in the natural and technical sciences"... or even "Drawing from the natural and technical sciences.." ?

Reviewer #2: As for my prior questions about classifying the issue sentiment of all tweets by the fastText algorithm, the authors need to elaborate more.

What’s more, the discussion is not well-established, which need to further revision.

Reviewer #3: (No Response)

7. PLOS authors have the option to publish the peer review history of their article (what does this mean?). If published, this will include your full peer review and any attached files.

Reviewer #1: No

Reviewer #2: **Yes: **HU Jieyi

Reviewer #3: No

---

## [Author Response · Author response to Decision Letter 1]

30 Jul 2021

Dear Barbara Guidi,

Thank you for inviting us to resubmit the paper upon suitable minor revisions and giving us the opportunity to address the issues raised by the reviewers to our revised version. We wish to thank the reviewers for their close reading of the revised version of their constructive comments. 

As specified in the decision letter, we have addressed the grammatical issues pointed out by reviewer 1, and this revised version of the article elaborates more details of the issue sentiment classification with the fastText algorithm, in accordance with the suggestions of reviewer 2. Additionally, we have uploaded an adjusted version of Figure 5, because we found a typo in this figure.

We hope you agree that our revision adequately addresses the concerns that the reviewers raised. 

Kind regards, 

the authors

---

## [Decision Letter · Decision Letter 2]

13 Aug 2021

Why it is important to consider negative ties when studying polarized debates: a signed network analysis of a Dutch cultural controversy on Twitter

PONE-D-21-08059R2

Dear Dr. Keuchenius,

We’re pleased to inform you that your manuscript has been judged scientifically suitable for publication and will be formally accepted for publication once it meets all outstanding technical requirements.

Kind regards,

Barbara Guidi

Academic Editor

PLOS ONE

Additional Editor Comments (optional):

Reviewers' comments:

Reviewer's Responses to Questions

**Comments to the Author**

1. If the authors have adequately addressed your comments raised in a previous round of review and you feel that this manuscript is now acceptable for publication, you may indicate that here to bypass the “Comments to the Author” section, enter your conflict of interest statement in the “Confidential to Editor” section, and submit your "Accept" recommendation.

Reviewer #1: All comments have been addressed

Reviewer #3: All comments have been addressed

2. Is the manuscript technically sound, and do the data support the conclusions?

Reviewer #1: (No Response)

Reviewer #3: Yes

3. Has the statistical analysis been performed appropriately and rigorously? 

Reviewer #1: (No Response)

Reviewer #3: Yes

4. Have the authors made all data underlying the findings in their manuscript fully available?

Reviewer #1: (No Response)

Reviewer #3: No

5. Is the manuscript presented in an intelligible fashion and written in standard English?

Reviewer #1: (No Response)

Reviewer #3: Yes

6. Review Comments to the Author

Reviewer #1: (No Response)

Reviewer #3: (No Response)

7. PLOS authors have the option to publish the peer review history of their article (what does this mean?). If published, this will include your full peer review and any attached files.

Reviewer #1: No

Reviewer #3: No

---

## [Editor Report · Acceptance letter]

17 Aug 2021

PONE-D-21-08059R2 

Why it is important to consider negative ties when studying polarized debates: a signed network analysis of a Dutch cultural controversy on Twitter 

Dear Dr. Keuchenius:

I'm pleased to inform you that your manuscript has been deemed suitable for publication in PLOS ONE. Congratulations! Your manuscript is now with our production department. 

Kind regards, 

on behalf of

Dr. Barbara Guidi 

Academic Editor

PLOS ONE